# Markers of Regenerative Processes in Patients with Bipolar Disorder: A Case-control Study

**DOI:** 10.3390/brainsci10070408

**Published:** 2020-06-30

**Authors:** Artur Reginia, Jerzy Samochowiec, Marcin Jabłoński, Ewa Ferensztajn-Rochowiak, Janusz K. Rybakowski, Arkadiusz Telesiński, Maciej Tarnowski, Błażej Misiak, Mariusz Z. Ratajczak, Jolanta Kucharska-Mazur

**Affiliations:** 1Department of Psychiatry, Pomeranian University of Medicine, 71-460 Szczecin, Poland; jerzysamochowiec@gmail.com (J.S.); marcinjablonski2@wp.pl (M.J.); jola_kucharska@tlen.pl (J.K.-M.); 2Department of Adult Psychiatry, Poznan University of Medical Sciences, 60-572 Poznań, Poland; ferensztajnewa@gmail.com (E.F.-R.); janusz.rybakowski@gmail.com (J.K.R.); 3Department of Psychiatric Nursing, Poznan University of Medical Sciences, 60-179 Poznań, Poland; 4Department of Bioengineering, Faculty of Environmental Management and Agriculture, West Pomeranian University of Technology, 71-434 Szczecin, Poland; atelesinski@zut.edu.pl; 5Chair and Department of Physiology, 70-111 Szczecin, Poland; maciej.tarnowski@pum.edu.pl; 6Department of Genetics, Wroclaw Medical University, 50-368 Wrocław, Poland; mblazej@interia.eu; 7Stem Cell Institute at James Graham Brown Cancer Center, University of Louisville, Louisville, KY 40202, USA; mariusz.ratajczak@louisville.edu; 8Department of Regenerative Medicine, Medical University of Warsaw, 02-097 Warsaw, Poland

**Keywords:** stem cells, VSEL, bipolar disorder

## Abstract

Progress in medical science has allowed the discovery of many factors affecting the pathogenesis of bipolar disorder, and among the most recent research directions are found regenerative and inflammatory processes. The role of regenerative processes remains particularly poorly explored, but available data encourage further research, which may explain the pathogenesis of bipolar disorder (BD). The aim of this study was to evaluate the mobilization of stem cells into peripheral blood, in patients with bipolar disorder during stable phase, not treated with lithium salts. The study included 30 unrelated individuals with the diagnosis of bipolar disorder, with disease duration of at least 10 years, not treated with lithium salts for at least five years prior to the study. The control group consisted of 30 healthy subjects, matched for age, sex, body mass index (BMI), origin, socio-demographic factors and nicotine use. Blood samples underwent cytometric analyses to assess concentrations of: Very Small Embryonic Like (VSEL) CD34+, VSEL AC133+, HSC CD34+, HSC AC133+. There were no significant differences in stem cell levels between patients with BD and healthy controls. However, the level of VSEL cells AC133 + was significantly higher in type I BD patients compared to healthy controls. Our results indicate a disturbance in regenerative processes in patients with bipolar disorder.

## 1. Introduction

Despite great efforts invested in understanding the pathophysiology of mental disorders, including bipolar disorder (BD), the exact mechanisms remain unclear [1]. Nevertheless, complex psychiatric disorders are essentially perceived in terms of neurological or systemic pathology [2]. Available models of mood disorders increasingly involve the role of local and systemic inflammatory processes as their molecular underpinnings [3,4]. Immune dysfunction is listed among the causative factors of degenerative changes within the central nervous system [5]. At the same time, the evidence suggests that inflammation may activate regenerative processes [6,7].

Stem cells constitute key elements of regenerative processes [8]. Relatively scarce in adults, they are located in niches that provide adequate conditions to preserve their properties, and maintain tissue homeostasis [9]. Their numbers increase in response to systemic or local inflammation, tissue damage or vigorous exercise [10]. Among many types of stem cells, Very Small Embryonic Like (VSEL) Stem Cells should be distinguished. Discovered and described relatively recently, they owe their name to their very small sizes — 3–6 μm in diameter and the presence of pluripotency markers, such as Oct4, Nanog, Rex-1 and SSEA-1 [11,12]. They are mobilized into peripheral blood in response to strenuous physical activity, infections, sepsis, myocardial infarction, stroke, burns, inflammatory bowel diseases and certain types of cancer [13,14,15]. There are also some reports of their potential role in the pathogenesis of psychotic disorders [16].

To participate in tissue regeneration, stem cells must be activated. Some of them undergo mobilization to the peripheral blood and reach the damaged areas [17,18], in response to a variety of factors, such as granulocyte-colony stimulating factor (G-CSF), stromal-derived factor-1 (SDF-1) or sphingosine-1 phosphate [19]. Recent evidence indicates that this process can also be activated by the components of the complement cascade [14,20]. We have recently demonstrated the effect of inflammatory processes on the course of BD, manifested by an increased concentration of complement components C3a and C5a in the peripheral blood [21]. However, there is a scarcity of studies investigating the regenerative processes and stem cells in patients with BD. Therefore, in this study, we aimed to assess the activation of regenerative processes in patients with BD, manifested in the mobilization of stem cells from bone marrow to the peripheral blood.

## 2. Materials and Methods

The study was approved by the Bioethical Committee of the Pomeranian Medical University at its meeting of 29.10.2014, resolution no. KB-0012/127/12, supplemented with the consents KB-0012/70/14 of 13.10.2014 and KB-0012/48/15 of 23.03.2015 and all participants gave written informed consent. Our sample included 30 unrelated BD patients with illness duration equal to or longer than 10 years, and 30 unrelated healthy controls, matched for age, sex, body mass index (BMI) and a frequency of cigarette smoking. Due to the potential effect of lithium on the mobilization of some stem cell populations, persons treated with lithium, within the period of five years prior to baseline were excluded from the study. Other exclusion criteria were as follows: Active substance dependence in the preceding six months (except for nicotine dependence), a history of severe organic brain damage, dementia-like cognitive impairment, severe somatic illnesses, glucose intolerance, active inflammatory disease (patients were excluded based on laboratory test results and physical examination), other mental disorders (except for personality disorders). At the time of the study, all participants were in euthymic mood. All patients met the criteria of BD in remission (F31.7) according to the 10th revision of the International Statistical Classification of Diseases and Related Health Problems (ICD-10) criteria.

The presence of mental disorders other than BD were excluded, using the MINI questionnaire [22]. To assess mood disorders, we used the Montgomery-Åsberg Depression Rating Scale (MADRS) [23] and the Young Mania Rating Scale (YMRS) [24]. To exclude suicidal tendencies, we used the Columbia Suicide Severity Rating Scale [25]. In addition, the anxiety level was assessed with the Hamilton Anxiety Scale [26,27]. Demographic data, family history, and history of presenting complaint were collected using a standardized medical history.

After expressing informed consent to participate in the study, each participant underwent a standard psychiatric examination, physical examination and psychometric assessment. The score of ≤10 points on the MADRS [28] and ≤ 12 points on the YMRS [29] indicated a euthymic state. Data on previous pharmacological treatment were retrieved from available medical records and interviews with the study participants. All patients were treated in accordance with the Polish guidelines for the pharmacological management of mood disorders [30]. Fasting venous blood was collected from the participants between 8:00 and 9:00 a.m., and immediately transferred for analysis.

At the time of the study, patients were receiving lamotrigine at doses of 50 to 300 mg/d (12 patients), valproic acid/sodium valproate at doses of 600 to 1500 mg/d (10 patients), quetiapine at doses of 100 to 700 mg/d (14 patients), olanzapine at doses of 5 to 20 mg/d (5 patients), clozapine at a dose of 225 mg/d (1 patient), and aripiprazole (1 patient), as their primary treatment. In addition, they were administered: sertraline (2 patients), venlafaxine (2 patients), escitalopram (1 patient), citalopram (1 patient), mirtazapine (1 patient), paroxetine (1 patient), clomipramine (1 patient), perazine (1 patient), levomepromazine (1 patient), chlorprothixene (1 patient). The medicine and its dose were determined by the attending physician. The research team did not modify the recommended treatment in any way.

To analyze the effect of treatment on the process of stem cell mobilization, doses of antipsychotics were converted into chlorpromazine equivalents [31,32,33]. Doses equivalent to 100mg/day of chlorpromazine were 7.5mg/day for aripiprazole, 120mg/day for chlorprothixene, 75mg/d for quetiapine, 50mg/day for clozapine, 100 mg/day for levomepromazine, 5mg/day for olanzapine, and 100mg/day for perazine.

### 2.1. Measurement of the Level of Stem Cells

Peripheral blood samples were lysed twice at room temperature for 10 minutes, using BD Pharm Lyse buffer (BD Bioscience), and then washed in phosphate buffered saline (PBS) supplemented with 2% fetal bovine serum (FBS; Sigma) to obtain the full number of nucleated cells (TNCs). These were then stained with markers of hematopoietic line cells using antibodies conjugated with fluorescein isothiocyanate (FITC) against human antigens: CD2 (RPA-2.10 clone); CD3 (UCHT1 clone); CD14 (M5E2 clone); CD16 (clone 3G8); CD19 (HIB19 clone); CD24 (ML5 clone); CD56 (NCAM16.2 clone); CD66b (G10F5 clone); and CD235a (GA-R2 clone) (all from BD Bioscience). The cells were also stained for a pan-leukocyte marker (CD45), phycoerythrin (PE) conjugated antibodies (clone HI30; BD Biosciences) and one of the following antigens: CD34: allophycocyanin conjugated antibodies (APC) (clone 581; BD Bioscience) or CD133 (CD133/1, APC-conjugated antibodies; Miltenyi Biotec). In addition, we used the following FITC conjugated isotype controls: mouse IgG1, ĸ (MOPC-21 clone), mouse IgG2a, (G155–178 clone), mouse IgG2b, ĸ (27–35 clone); PE-conjugated IgG1, ĸ (MOPC-21 clone) and APC-conjugated mouse IgG1, ĸ (MOPC-21 clone) (all from BD Bioscience). We also used APC-conjugated mouse IgG1 antibodies (IS5-21F5 clone; Miltenyi Biotec). Staining was performed in buffered salt (PBS) with 2% FBS, on ice, for 30 minutes. The cells were then washed, re-suspended in liquid and analyzed using a NAVIOS flow cytometer (Beckman Coulter). At least 106 events were obtained for each sample. The total VSEL and hematopoietic stem cell (HSC) cell counts were calculated (individually for each patient) per 1 μL of peripheral blood, based on the percentage of these cells found cytometrically and the total number of white blood cells in 1 μL of peripheral blood. The Kaluza software (Beckman Coulter) was used for the analysis.

### 2.2. Data Analysis

The distribution of categorical variables was compared using the chi2 test. The Shaprio-Wilk test was used to assess normality of data distribution. Due to non-normal distribution of continuous variables, the Mann-Whitney U test was used. All data are presented as the mean ± SD. The Spearman rank correlation coefficients were used to assess the relationship between continuous variables. The analysis of co-variance (ANCOVA) was performed to test between-group differences in the levels of stem cells after co-varying for age, sex, BMI and the dosage of various medications. In case of non-normal distribution of the levels of stem cells, logarithmic transformation was performed. Statistical significance was set at *p* < 0.05. Data were analyzed with the Statistical Package for Social Sciences, version 20 (SPSS Inc., Chicago, IL, USA).

## 3. Results

General characteristics of patients and healthy controls were shown in Table 1. There were no significant between-group differences in age, sex, BMI and cigarette smoking status. The majority of patients had type I BD.

There were no significant differences in stem cell levels between patients with BD and healthy controls (Table 2). However, the level of VSEL cells (Lin−/CD45−/AC133 +) was significantly higher in type I BD patients compared to healthy controls. This difference remained significant (F = 4.80, *p* = 0.034) after co-varying for age (F = 0.11, *p* = 0.742), sex (F < 0.001, *p* = 0.961), BMI (F = 0.92, *p* = 0.344), chlorpromazine equivalent dosage (F = 0.83, p = 0.367), the dosage of valproic acid/sodium valproate (F = 2.78, *p* = 0.103) and the dosage of lamotrigine (F = 0.19, *p* = 0.662). No significant difference in stem cell counts was found between Type I and type II BD patients.

There were no significant correlations between the dosage of medications and the levels of stem cells (Table 3).

## 4. Discussion

BD is a disease of very complex course and etiology. It seems that depending on BDs phase, there are some different kinds of pathologies affecting brain function. This fact has been indirectly proven by various levels of cytokines in different mood phases [34]. Therefore, while planning the first research in the area of regenerative medicine, we have decided to estimate the long-term effects of all these processes by choosing subjects with euthymia who suffered from BD for more than 10 years. It should be noted that lithium may exert a significant, though not fully explained effect on various types of stem cells [35]. For this reason, we excluded patients treated with lithium salts to avoid its effect on mobilization of examined stem cells.

In our study, we found no differences in VSEL or HSC stem cell concentrations between the entire study population and control group, or BD I and II patients. There was a difference in VSEL (Lin−/CD45−/AC133+) concentration between BD I patients and the controls. However, further studies on a larger group of patients are necessary. This last result seems to be very promising because researches increasingly often prove differences in etiology of BD I and BD II [36,37]. The “continuum” theory of mental disorders suggests the existence of a common basis for psychoses, i.e., schizophrenia and BD, BD I in particular, as evidenced by genetic data [38,39] and the concentrations of inflammatory factors [40]. The differences between the mobilization of VSEL (Lin−/CD45−/AC133+) in BD I and BD II observed in this study seem to support this way of thinking. The observed mobilization of VSEL(Lin−/CD45−/AC133+) in BD I, as a BD variant with a more severe course that is etiologically closer to schizophrenia, seems to be the effect of the exacerbation of inflammatory processes influencing the mobilization of stem cells in these patients when compared to BD II. To date, there have been relatively few reports on the dynamics of stem cells, especially VSEL cells, in patients with mental disorders. The only available data on the behavior of VSELs in BD, relate to patients treated with lithium salts. It has been shown that by reducing the numbers of VSELs circulating in peripheral blood, long-term lithium treatment may suppress the activation of regenerative processes, including perhaps those with a pathological course [35]. That is also evidenced by the fact that the concentration of VSELs (CD34+) in the subgroup of 15 patients with BD not treated with lithium salts was higher in comparison to healthy subjects [41].

It is also worth to consider the issue in the context of the first research concerning behavior of VSELs in first episode of psychosis (FEP) performed by Kucharska-Mazur [16], who published their findings on VSELs and HSCs in first-episode psychosis, demonstrating higher numbers of VSEL (Lin−/CD45−/CD34+) in patients with FEP both before and after antipsychotic treatment. Elevated VSEL (Lin−/CD45−/CD34+) counts indicated a disturbance within regenerative processes. No such associations were observed for VSEL (Lin−/CD45−/AC133+), which was different compared to our study.

Unfortunately, there is no data about VSELs in patients with depressive disorders, but subsequent reports on the potential role of stem cells have been performed in patients with panic disorder. In this group of patients, a reduced number of HSCs (Lin−/CD45+/AC133+) was observed. Interestingly, low HSC concentration were not significantly affected by the administered short-term antidepressant therapy. Other investigated cell counts, i.e., VSEL and HSC (Lin−/CD45+/CD34+) did not differ significantly between patients and healthy controls [42].

There is no doubt that inflammatory reactions are involved in the pathogenesis of BD. Immune dysfunctions can take the form of inflammatory changes in the CNS, and, even more frequently, a generalized inflammatory response [43,44,45,46]. A very compelling concept of “sterile inflammation” supports evidence that inflammatory responses are initiated directly in brain tissue of the CNS, triggered by prolonged or chronic activation of the complement cascade via the MBL-MASP pathway [47]. Given the chronic, relapsing/remitting course of BD, the activation of immune processes may also be linked to the phase of BD [48]. In our study, all participants were euthymic, had a ≥ 10-year history of BD, and their stem cell count and concentrations of factors affecting their mobilization were calculated from the peripheral blood, which suggests that what we determined were in fact markers of chronic inflammatory response.

Therefore, in spite of the observed elevated levels of complement components C3a and C5a, considered factors responsible for stem cell mobilization to peripheral blood [21], it seems that there is no adequate increase in the numbers of VSELs or HSCs in peripheral blood of BD patients in remission (euthymic mood), with a ≥ 10 year illness duration. Therefore, the elevated concentrations of factors responsible for stem cell movement do not seem to be linked with their mobilization from bone marrow to peripheral blood. However, it should be noted that increased stem cell mobilization may occur during periods of exacerbation, especially since the course of BD may include psychotic symptoms [49]. This issue, however, requires further study. In the periods of exacerbation, there is also evidence of transient disruption of the blood-brain barrier in BD patients, which may facilitate the “communication” of nerve tissue with peripheral processes [50].

Our study has certain limitations that need to be raised. Firstly, our sample was not large and thus type II error cannot be excluded. Another important point is that all patients were euthymic. Therefore, we were unable to address whether altered levels of stem cells are related to any specific mood phase. Similarly, we were not able to assess the association with psychopathological manifestation. Moreover, a lack of a longitudinal study design does not allow to establish causal associations. Finally, although we controlled for the dosage of various medications, the use of mood stabilizers and antipsychotics accounts for the alterations observed in our study cannot be ruled out.

## 5. Conclusions

Although, the exact pathomechanism of changes in the regenerative system in mental illness is not fully understood, our results indicate a disturbance in regenerative processes in patients with BD. The behavior of stem cells in BD differs from that in FEP and anxiety disorders. It also differs depending on the treatment with lithium salts and other medicaments. This topic requires research on stem cells in peripheral blood in other mental disorders, especially depressive disorders, and also on BD subjects during different phases of this disorder. This field seems to be full of promise in searching for biomarkers helpful in the differential diagnostics procedure of mental disorders.

## Figures and Tables

**Table 1 brainsci-10-00408-t001:** Clinical and demographic characteristics of the patient and control groups.

	BD Patients *n* = 30	Healthy Controls *n* = 30	*p*
Age, Years	48.08 ± 11.54	43.90 ± 10.74	0.070
Sex, M/F (%)	15 (50.0)/15 (50.0)	13 (43.3)/17 (56.7)	0.665
Smoking, *n* (%)	13 (43.3)	9 (40.9)	0.890
BMI, kg/m^2^	26.53 ± 4.86	25.15 ± 4.48	0.217
Type I BD, *n* (%)	22 (73.3%)	-	-
Illness Duration, Years	17.63 ± 8.22	-	-
Treatment Duration, Years	11.84 ± 8.73	-	-
MADRS	0.07 ± 1.71	-	-
YMRS	0.93 ± 1.26	-	-
HAM-A	1.07 ± 2.30	-	-

**Table 2 brainsci-10-00408-t002:** The levels of stem cells in patients with BD and healthy controls.

**SCs**	**BD Patients (mean ± SD)**	**Healthy Controls (mean ± SD)**	***p***
VSEL (Lin−/CD45−/CD34+)	0.135 ± 0.112	0.148 ± 0.080	0.446
HSC (Lin−/CD45+/CD34+)	1.139 ± 0.567	1.148 ± 0.730	0.511
VSEL (Lin−/CD45−/AC133+)	0.129 ± 0.132	0.084 ± 0.065	0.134
HSC (Lin−/CD45+/AC133+)	0.973 ± 0.547	1.143 ± 0.754	0.673
SCs	BD I Patients (mean ± SD)	Healthy Controls (mean ± SD)	*p*
VSEL (Lin−/CD45−/CD34+)	0.148 ± 0.099	0.1477 ± 0.080	0.456
HSC (Lin−/CD45+/CD34+)	1.173 ± 0.612	1.1476 ± 0.730	0.502
VSEL (Lin−/CD45−/AC133+)	0.137 0.133	0.0840 ± 0.065	0.029
HSC (Lin−/CD45+/AC133+)	1.004 ± 0.601	1.1432 ± 0.754	0.613
SCs	BD II Patients (mean ± SD)	Healthy Controls (mean ± SD)	*p*
VSEL (Lin−/CD45−/CD34+)	0.170 ± 0.1710	0.1247 ± 0.080	0.712
HSC (Lin−/CD45+/CD34+)	1.044 ± 0.440	1.1476 ± 0.730	0.792
VSEL (Lin−/CD45−/AC133+)	0.109 ± 0.136	0.0840 ± 0.065	0.586
HSC (Lin−/CD45+/AC133+)	0.894 ± 0.383	1.1432 ± 0.754	0.986

**Table 3 brainsci-10-00408-t003:** Correlation between drug dose and stem cell count.

	Doses of Antipsychotics Equivalent to 100mg/day of Chlorpromazine		Valproic Acid/Sodium Valproate		Lamotrigine	
	*R_s_*	*p*	*R_s_*	*p*	*R_s_*	*p*
VSEL (Lin−/CD45−/CD34+)	0.0936	0.6947	0.3227	0.2408	0.1785	0.5415
HSC (Lin−/CD45+/CD34+)	0.1253	0.5987	−0.191	0.6723	0.1428	0.6263
VSEL (Lin−/CD45−/AC133+)	0.0626	0.7930	0.1829	0.6328	0.2071	0.4775
HSC (Lin−/CD45+/AC133+)	0.1479	0.5337	−0.1037	0.7130	−0.1160	0.6913

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
