# Peer review of "Markers of Regenerative Processes in Patients with Bipolar Disorder: A Case-control Study"

_brainsci, 2020, doi:10.3390/brainsci10070408_

Round 1

Reviewer 1 Report

The manuscript number brainsci-832043 highlights potential biomarkers of regenerative processes that could be affected in BD patients. This study is interesting, well documented and reports clear results including main limitations which authors have been taking into consideration.

Despite of no significant differences found in BD patients compared to healthy controls, the level of VSEL cells AC133+ was increased in type I BD. These results are particularly interesting in this “euthymic” phase of BD and point to know possible changes in regenerative processes in other phases of this disorder by stem cells evaluation.

However, some minor points could be further addressed:

  • Regarding methods for measurement of the level of stem cells (section 2.1.), VSEL and HSC cell counts were calculated in 1 ml of peripheral blood (lines 129-132), however, table 2 indicates that this volume is 1 µ Please, authors must clarify this issue.
  • It would be convenient to add a sentence in 2.2. Data analysis section including if all data are presented as the mean ± SD or SEM.
  • In the results section, line 150, please replace table 1 by table 2.
  • It would be convenient to include a graph panel with flow cytometry graphs for VSEL and HSC cells. These graphical representations could contribute to enrich results section.
  • In table 2, column titles about stem cell counts could be simplified as BD patients and healthy controls and BD I or II patients and healthy controls.
  • References format in the manuscript must be revised, there are at least two cited by no numbered format (lines 61 and 214-215).
  • Abbreviations should be defined at first mention and used consistently thereafter, please authors should be revised them, it is the case of ICD-10, HSC or FEP.
  • Please authors must revise remove extra spacing between words along the manuscript and table 2.
  • Finally, how authors could explain differences found in VSEL (Lin-/CD45-/AC133+) cells in BD type I patients but not in type II compared to healthy controls? This point must be discussed in the discussion section.

Author Response

I would like to thank you for your comments and time spent on the manuscript evaluation. Thank you for giving us the opportunity to revise and improve our article. We've listed all the changes below

  • Cell counts were calculated in 1 μl of peripheral blood. This has been corrected in lines 129-132.
  • All data are presented as the mean ± SD. It has been noted in section “Data analysis”, line 137.
  • In the results section, line 150 table 1 has been replaced by table 2.
  • Stem cell concentrations were determined for us by a separate The results were provided to us in the form of a tables but not cytometry graphs.
  • Table 2 column titles had been simplified according to your suggestion.
  • References had been revised and corrected in lines 61 and 224. Three new references have been added in line 180. These are positions 38, 39 and 40.
  • Abbreviations have been defined and used consistently thereafter.
  • 10th revision of the International Statistical Classification of Diseases and Related Health Problems (ICD-10) (lines 82-83)
  • hematopoietic stem cell (HSC) (lines 129- 130)
  • first episode of psychosis (FEP) (line 193)
  • using of the abbreviation of BD and bipolar disorder has been also corrected

  • We have also added explanation of mobilization of VSEL(Lin-/CD45-/AC133+) in BD I patients but not BD II which is below:

The „continuum” theory of mental disorders suggests the existence of a common basis for psychoses, i.e. schizophrenia and bipolar disorders, BD I in particular, as evidenced by genetic data and the concentrations of inflammatory factors. The differences between the mobilization of VSEL(Lin-/CD45-/AC133+) in BD I and BD II observed in this study seem to support this way of thinking. The observed mobilization of VSEL(Lin-/CD45-/AC133+) in BD I, as a BD variant with a more severe course that is etiologically closer to schizophrenia, seems to be the effect of the exacerbation of inflammatory processes influencing the mobilization of stem cells in these patients when compared to BD II.

Reviewer 2 Report

The research design and implementation are sound, but I agree that this needs to be repeated on a significantly larger population. The VSEL stem cell line that was increased in the BD patients needs more analysis - is it an inflammation response, as noted, or something else? Overall the study is well done and needs only a few spell checks. One thing I would like to see is some visuals regarding the cell types involved - I think that would help with presentation of results.

Author Response

I would like to thank you for your comments and time spent on the manuscript evaluation. Thank you for giving us the opportunity to revise and improve our article.

All spell checks have been made.

The VSELS are mobilized to peripheral blood as result of inflammatory reactions. We have added explanation of mobilization of VSEL in BD I but not BD II.

The „continuum” theory of mental disorders suggests the existence of a common basis for psychoses, i.e. schizophrenia and bipolar disorders, BD I in particular, as evidenced by genetic data and the concentrations of inflammatory factors. The differences between the mobilization of VSEL(Lin-/CD45-/AC133+) in BD I and BD II observed in this study seem to support this way of thinking. The observed mobilization of VSEL(Lin-/CD45-/AC133+) in BD I, as a BD variant with a more severe course that is etiologically closer to schizophrenia, seems to be the effect of the exacerbation of inflammatory processes influencing the mobilization of stem cells in these patients when compared to BD II.

You have asked us about "some visuals regarding the cell types involved". Could you specify to us what kind of visuals you would like to see? Did I understand correctly that it was a graphic presentation of the stem cell mobilization process?